# The Mycotoxin De-Epoxy-Deoxynivalenol (DOM-1) Increases Endoplasmic Reticulum Stress in Ovarian Theca Cells

**DOI:** 10.3390/toxins15030228

**Published:** 2023-03-17

**Authors:** Angelica D. Reyes-Perea, Hilda M. Guerrero-Netro, Europa Meza-Serrano, Anthony Estienne, Christopher A. Price

**Affiliations:** 1Centre de Recherche en Reproduction et Fertilité (CRRF), Faculté de Médecine Vétérinaire, Université de Montréal, St-Hyacinthe, QC J2S 2M2, Canada; 2Departamento de Reproducción, Facultad de Medicina Veterinaria y Zootecnia, Universidad Nacional Autonoma de Mexico, Mexico City 04510, Mexico

**Keywords:** deoxynivalenol, follicle, cattle, steroidogenesis, autophagy

## Abstract

Deoxynivalenol (DON) is a major mycotoxin present in animal feed and negatively affects growth and reproduction in farm species, including pigs and cattle. The mechanism of DON action involves the ribotoxic stress response (RSR), and it acts directly on ovarian granulosa cells to increase cell death. In ruminants, DON is metabolized to de-epoxy-DON (DOM-1), which cannot activate the RSR but has been shown to increase cell death in ovarian theca cells. In the present study, we determined if DOM-1 acts on bovine theca cells through endoplasmic stress using an established serum-free cell culture model and to assess whether also DON activates endoplasmic stress in granulosa cells. The results show that DOM-1 increased the cleavage of ATF6 protein, increased the phosphorylation of EIF2AK3, and increased the abundance of cleaved *XBP1* mRNA. Activation of these pathways led to an increased abundance of mRNA of the ER stress target genes *GRP78*, *GRP94*, and *CHOP*. Although CHOP is widely associated with autophagy, inhibition of autophagy did not alter the response of theca cells to DOM-1. The addition of DON to granulosa cells partially increased ER stress pathways but failed to increase the abundance of mRNA of ER stress target genes. We conclude that the mechanism of action of DOM-1, at least in bovine theca cells, is through the activation of ER stress.

## 1. Introduction

Deoxynivalenol (DON), produced by Fusarium species, is a B-trichothecene that is reported as occurring with high frequency in animal feeds [1,2,3]. In vivo exposure to DON negatively impacts feed intake, growth, and intestinal and immune health, especially in pigs [4,5,6]. Exposure of pigs to Fusarium toxins in vivo causes degeneration of oocytes [7], and in vitro DON has been shown to impair oocyte quality and steroidogenesis in ovarian granulosa cells in pigs [8,9,10] and decrease granulosa cell steroidogenesis and increase apoptosis in bovine granulosa cell models [11,12].

The mechanism of action of DON is widely accepted to involve the ribotoxic stress response (RSR), in which DON interacts with ribosomal RNA and phosphorylates eukaryotic translation initiation factor 2 alpha kinase 2 (EIF2AK2; formally known as PKR) and Src family tyrosine kinase (HCK). These then phosphorylate eukaryotic translation initiation factor alpha (EIF2A), which leads to the phosphorylation of mitogen-activated protein kinases (MAPK), including MAPK3/1, MAPK14 (p38) and MAPK8 (JNK), resulting in apoptosis and cell death [13,14,15]. Exposure of bovine granulosa cells to DON rapidly increased the phosphorylation of MAPK3/1, MAPK14, and MAPK8 [11], suggesting that the RSR was activated in this cell type.

Ruminants are resistant to the toxic effects of DON as most ingested DON is metabolized in the rumen to de-epoxy-DON (DOM-1) [16], which is considerably less toxic than DON and does not activate the RSR [17,18,19]. However, DOM-1 is not inert; it has been shown to increase the immune response in pigs in vivo [20], to decrease follicle development in vivo in cattle [21], and to increase apoptosis of bovine ovarian theca cells in vitro [22]. The mechanism of action of DOM-1 is unclear, as although it stimulated EIF2AK2 and MAPK3/1 phosphorylation in theca cells, it also stimulated the expression of the endoplasmic reticulum stress marker ATF4 [22].

Endoplasmic reticulum (ER) stress is typically activated by the accumulation of unfolded proteins in the ER, leading to apoptosis and autophagy. Activation of ER stress involves the degradation of the chaperone GRP78 in the ER lumen, which then permits the activation of three ER stress pathways: (a) activation of Endoplasmic reticulum to nucleus signaling 1 (ERN1, also known as IRE1a) and consequent cleavage of *XBP1* mRNA; (b) phosphorylation of EIF2AK3 (also known as PERK); and (c) cleavage of membrane-bound Activated transcription factor 6 (ATF6) and release into the cytoplasm [23]. Cleaved XBP1 and ATF6 are transcription factors, whereas EIF2AK3 phosphorylates EIF2A and increases expression of ATF4, which in turn results in expression of beclin-1 (BECN1), activation of LC3 protein (encoded by the *MAP1LC3B* gene); these pathways may culminate in increased autophagy [24,25].

There is some evidence that DON can activate ER stress pathways, as it increased degradation of GRP78 protein and stimulated cleavage of *XBP1* mRNA in mouse macrophages [26], and increased *ATF6* mRNA abundance in human lymphocytes [27] and bovine mammary epithelial cells [28]. DON has also been reported to increase autophagy in the mouse spleen and porcine intestinal epithelial cells, among others [29,30], and autophagy has been proposed as a protective mechanism against DON toxicity [31].

In the ovary, DON increased granulosa cell apoptosis associated with MAPK3/1, MAPK8, and MAPK14 phosphorylation and increased expression of the ‘death ligand’, FASLG [11]. In theca cells, these same markers are increased by DOM-1 but not by DON [22]. It is unknown if these mycotoxins increase autophagy in the ovary or whether ER stress is activated. Therefore, we hypothesized that the mechanism of action of DOM-1 in the ovary includes ER stress and autophagy. The objective of the current study was to determine the role of autophagy and to identify which ER stress pathways are activated in theca cells in response to DOM-1. As a comparison, we also included experiments applying DON to granulosa cells, which is known to increase apoptosis through the RSR [11].

## 2. Results

To determine whether mycotoxins increase autophagy in the ovary, we first measured the abundance of LC3 and BECN1 mRNA/protein. Culture of granulosa cells with DON significantly increased *MAP1LC3B* and *BECN1* mRNA abundance, whereas the culture of theca cells with DOM-1 increased *MAP1LC3B* but not *BECN1* mRNA abundance (Figure 1).

Total LC3 protein abundance was assessed by immunofluorescence; DON increased protein abundance in granulosa cells, and DOM-1 increased protein abundance in theca cells after 12 h of exposure (Figure 2A). The relative abundance of intact (LC3-I) and cleaved (LC3-II) MAP1LC3B protein was then measured by immunoblotting. In granulosa cells, DON increased LC3-II but not LC3-I protein abundance at 12 h of exposure, whereas in theca cells, DOM-1 increased the abundance of LC3-II protein at 12 and 24 h exposure; LC3-I abundance was also increased by DOM-1 at 24 h of exposure (Figure 2B).

The functional relevance of autophagy was assessed by pharmacological inhibition of autophagosome formation with the ULK1 inhibitor, XST-14. In granulosa cells, the addition of inhibitor alone did not alter the rate of apoptosis, and the addition of DON significantly increased the rate of apoptosis (Figure 3A); cotreatment of granulosa cells with DON and inhibitor increased the proportion of apoptotic cells compared to DON alone. The addition of DON alone increased the abundance of mRNA encoding the apoptosis-related proteins BAX, BCL2, GADD45B, and BID, and the pro-autophagic proteins MAP1LC3B and GRP78. Cotreatment with DON plus ULK1 inhibitor further increased the abundance of mRNA encoding BAX, BCL2, GADD45B, and BID compared with DON alone (Figure 3B), suggesting that autophagy is a protective response to DON.

The addition of DOM-1 to theca cells also increased the rate of apoptosis (Figure 3A); however, cotreatment with DOM-1 and inhibitor did not alter the rate of apoptosis caused by DOM-1 alone. The addition of DOM-1 alone to theca cells increased the abundance of mRNA encoding MAP1LC3B, BAX, BCL2, GADD45B, and BID, whereas cotreatment with DOM-1 and inhibitor did not alter most apoptosis/autophagy markers compared with DON alone, but the decreased abundance of mRNA encoding the DNA damage/repair protein GADD45B (Figure 3B).

Both DON and DOM-1 increased the abundance of mRNA encoding the ER chaperone GRP78 (Figure 3B), and inhibition of autophagy further increased *GRP78* mRNA abundance in theca cells treated with DOM-1 but not in granulosa cells treated with DON. This suggests that autophagy induced by DOM-1 involves ER stress. We then determined which of the three main ER stress pathways are activated by DON and DOM-1. In granulosa cells, DON caused a transient increase in cleaved ATF6 protein abundance, a sustained increase in EIF2AK3 (PERK) phosphorylation over 12–24 h post-treatment, and a modest increase in ERN1 protein abundance at 24 h treatment (Figure 4A). This was accompanied by an increased abundance of mRNA encoding CHOP but not cleaved XBP1 at 8 h of treatment (Figure 4B), and decreased abundance of mRNA encoding GRP94, another ER protein involved in autophagy.

In theca cells, the addition of DOM-1 caused an increase in cleaved ATF6 abundance over 8–12 h post-treatment, and an increase in EIF2AK3 phosphorylation over 12–24 h post-treatment. The abundance of ERN1 protein abundance was increased at 8 h of treatment and returned to pretreatment levels by 12 h. Treatment with DOM-1 increased the abundance of mRNA encoding CHOP, cleaved XBP1, GRP87, and GRP94 at 8 h of treatment (Figure 5).

## 3. Discussion

The mycotoxin metabolite DOM-1 is generally considered to be relatively inert but has been shown to be active in the ovary and intestine [20,21,22]. As DOM-1 lacks the epoxy group necessary for initiating the RSR, the mechanism of action of DOM-1 was unclear. In this study, we show that DOM-1 activates the ER stress response as evidenced by increased cleavage of ATF6, increased phosphorylation of EIF2AK3, and increased abundance of cleaved *XBP1* mRNA.

Although there is likely cross-talk between the arms of the ER stress response, and between ER stress and the RSR, the present results argue for the activation of all three ER stress pathways by DOM-1 in theca cells. Firstly, the clear increase in ATF6 cleavage was accompanied by an increased abundance of mRNA of ATF6 target genes, including those encoding the ER-resident proteins GRP78 and GRP94 [32]. The second arm involves ERN1-XBP1 signaling, and DOM-1 increased in total ERN1 protein abundance and increased abundance of cleaved *XBP1* mRNA, which is an indirect measurement of ERN1 activity. Increased abundance of cleaved *XBP1* mRNA also leads to increased *GRP78* and *GRP94* mRNA abundance, depending on cell type, and it is likely that the ATF6 and ERN1-XBP1 pathways are functionally redundant [33]. Thirdly, DOM-1 increased the phosphorylation of EIF2AK3 and increased the abundance of mRNA of the EIF2A target CHOP.

When applied to bovine granulosa cells, DON also activated some ER stress pathways but not to the same extent as DOM-1 in theca cells. We observed an increase in cleaved ATF6 protein abundance in response to DON, but no expected increase in *GRP94* mRNA abundance. There was an increase in total ERN1 protein abundance, but this was not accompanied by an increase in cleaved *XBP1* mRNA abundance. Finally, the addition of DON increased phospho-EIF2AK3 and consequently *CHOP* mRNA abundance, although the increase in *CHOP* mRNA abundance could equally likely be affected through the RSR and EIF2AK2 (PKR) activation [13,14,15]. Thus, the data suggest that ER stress is not a likely direct target of DON action in granulosa cells.

All arms of the ER stress response have been involved in promoting autophagy in various cell types (reviewed in [34]). Increased expression of ATF4 and CHOP can stimulate BECN1 and MAP1LC3B accumulation, cleaved *XBP1* mRNA encodes a protein (XBP1s) that can activate *BECN1* gene transcription, and cleaved ATF6 may induce autophagy by inhibiting AKT phosphorylation or indirectly activating BECN1. These pathways then lead to the cleavage of LC3 protein and autophagosome formation. Although DOM-1 increased cleavage of LC3 in theca cells in the present study, there was no increase in *BECN1* mRNA abundance and no effect of inhibiting autophagy on DOM-1-dependent apoptosis or gene expression, suggesting that autophagy was not being activated by this metabolite in theca cells at the dose used. In contrast, DON increased *BECN1* mRNA abundance in granulosa cells, and inhibition of autophagy increased apoptosis and abundance of mRNA encoding apoptosis-related proteins, suggesting that autophagy is a protective mechanism activated in response to DON in the ovary, as has been suggested for rat neurons [31].

Although we have investigated the effect of DON and of DOM-1 on ER stress pathways in ovarian follicle cells, we cannot directly compare the mechanisms of action of each mycotoxin owing to the different cell types and doses used. Theca cells appear particularly sensitive to DOM-1 as 1 ng/mL of this metabolite impacts thecal steroidogenesis, but 100 ng/mL is required to affect granulosa cell steroidogenesis [22]. Whether lower doses of DON would affect the ER stress response in granulosa cells independently of the RSR is unknown, but lower doses do not appear to affect granulosa cell function in cattle [11,12].

## 4. Conclusions

The present data demonstrate that DOM-1, the non-toxic metabolite of DON, activates endoplasmic reticulum stress in bovine theca cells, and that this likely explains in part the observed effect of DOM-1 on theca cell growth and function [22].

## 5. Materials and Methods

### 5.1. Granulosa Cell Culture

All materials were obtained from Life Technologies Inc. (Burlington, ON, Canada). Bovine granulosa cells were cultured in a serum-free medium and maintained estradiol and progesterone secretion and responsiveness to FSH. Bovine ovaries from adult cows were obtained from a local abattoir, independently of the stage of the estrous cycle, and carried to the laboratory in phosphate-buffered saline (PBS; 30 °C) with penicillin (100 IU) and streptomycin (100 μg/mL). Follicles between 2–5 mm in diameter were isolated, and granulosa cells were removed by gently scraping the wall into the culture medium. The cell suspension was filtered through a 150 μm mesh steel sieve (Sigma-Aldrich Canada, Oakville, ON, Canada), and cell viability was determined by Trypan blue dye exclusion. Cells were placed into 24-well tissue plates (Sarstedt Inc., Newton, NC, USA) at the density of 500,000 viable cells in 500 μL DMEM/F12 with 25 mM HEPES, sodium bicarbonate (10 mM), bovine serum albumin (BSA)(0.1%; Sigma-Aldrich), penicillin (100 U/mL), sodium selenite (4 ng/mL), androstenedione (10^−6^ M), and bovine FSH (1 ng/mL starting on day 2, AFP5346D; National Hormone and Peptide Program, Torrance, CA, USA). Cells were cultures at 37 °C in 5% CO_2_, 95% air for up to 6 days.

### 5.2. Theca Cell Culture

Theca layers were removed from the stroma of 4–6 mm diameter follicles and incubated with collagenase type IV (1 mg/mL; Sigma-Aldrich) and trypsin inhibitor (100 μg/mL; Sigma-Aldrich) as previously described [35]. The cell suspension was filtered through a 150 μm mesh steel sieve and centrifuged (10 min at 800× *g*). Cell pellets were resuspended in PBS and then subjected to an osmotic shock to destroy red blood cells. Cell viability was determined by Trypan blue dye exclusion. Cells were cultured in McCoy 5A medium with the addition of 10 ng/mL bovine insulin, 10 mM HEPES, 100 IU/mL penicillin, 1 μg/mL fungizone, 5 μg/mL transferrin, 2 mM L-glutamine, 0.1% BSA, 5 ng/mL sodium selenite, and 0.8 ng/mL bovine LH (AFP5551B; NIDDK) in 24-well tissue plates (Sarstedt Inc. Newton, NC, USA) at a density of 250,000 viable cells/mL medium at 37 °C in 5% CO_2_, 95% air for up to 6 days with medium changes on days 2 and 4. Under these conditions, theca cells are responsive to LH and maintain testosterone and progesterone secretion.

### 5.3. Experimental Design

Biopure grade DON and DOM-1 (Romer Labs, Newark, DE, USA) were reconstituted in methanol for experimental treatments. On day 5 of culture, DON was added to granulosa cells at 100 ng/mL, which is known to inhibit cell function and activate MAPK signaling in this cell type [11]. Theca cells were similarly exposed to DOM-1 but at 1 ng/mL, a dose that inhibits cell function and activates MAPK signaling [22]. The autophagy inhibitor, XST-14 (HY-137506, MedChem Express, Monmouth Junction, NJ, USA), was diluted in DMSO and added to cells on day 5 of culture at a final concentration of 5 μM for 60 min prior to mycotoxin treatment. Controls received the vehicle alone.

### 5.4. Total RNA Extraction and Real-Time PCR

Total RNA was isolated from cells with the RNA Mini Kit (Qiagen, Hilden, Germany), and RNA was quantified by absorbance at 260 nm. Complementary DNA was synthesized from 200 ng total RNA with SuperScript^®^ VILO cDNA Synthesis Kit. Real-time PCR was performed with 2× Power SYBR Green PCR Master Mix in a volume of 15 μL with a CFX-96 Real-Time PCR Detection System (Bio-Rad Laboratories Ltd., Hercules, CA, USA). The bovine-specific primers are presented in Table 1. Reactions were performed with an initial heating to 95 °C for 3 min, followed by 40 cycles of 15 s at 95 °C, 30 s at 60 °C, and 30 s at 72 °C. Amplification of single products was verified by melting curve analysis. Samples were run in triplicate, and abundance was calculated relative to histone H2AFZ and beta-actin, which are stable across the treatments reported here. The coefficients of variation for CT values were between 0.3 and 1.5%. Data were normalized with the 2^−△△Ct^ method, including correction for amplification efficiency.

### 5.5. Immunoblotting

Theca or granulosa cells were lysed in 100 μL/well RIPA buffer (25 mM Tris–HCl pH 7.6, 150 mM NaCl, 1% sodium deoxycholate, 1% NP-40, 0.1% SDS, and phosphatase and protease inhibitor cocktails). Protein concentrations were measured by BCA protein assay (Pierce, Rockford, IL, USA). Proteins were resolved through 10% SDS–PAGE gels (3 μg/lane) and transferred onto PVDF membranes in a Bio-Rad wet Blot Transfer Cell apparatus for 2 h (transfer buffer: 48 mM Tris-base, 39 mM glycine, 20% methanol, pH 8.3). After transfer, membranes were blocked in either in TTBS (10 mM Tris–HCl, 150 mM NaCl, 0.1% Tween-20, pH 7.5) or 5% skim milk in TTBS for 1 h and incubated with the first antibody as described in Table 2. Membranes were then washed in TTBS and incubated with 1:10,000 HRP-conjugated anti-rabbit IgG (GE Healthcare Canada at room temperature for 1 h). Membranes were washed and reprobed with an antibody raised against β-actin (ACTB) as a loading control, whereas blots for ATF6 (50 kDa) were stripped to be re-incubated with ACTB as the proteins are of similar size. Protein bands were revealed by chemiluminescence (ECL, Bio-Rad Laboratories Ltd., Hercules, CA, USA), and band density was analyzed with Image Lab software (Bio-Rad).

### 5.6. Immunofluorescence

Cultured cells were fixed for 20 min in 4% paraformaldehyde, washed first in 2% Triton-X and then in 0.05% Tween, incubated in 5% BSA to block non-specific binding sites, and then incubated with the LC3 antibody (1:150). Fixed cells were then washed in PBS, and Cy3-conjugated second antibody (Jackson ImmunoResearch, West Grove, PA, USA) was applied, followed by counterstaining with 4′,6-diamidino-2-phenylidole (DAPI). An epifluorescence microscope (Zeiss, Jena, Germany) was used to capture digital images, and fluorescence intensities of Cy3 and DAPI were quantified in each field with ImageJ software (NIH). Results are expressed relative to the intensity of DAPI fluorescence.

### 5.7. Flow Cytometry

Flow cytometry was used to measure the apoptosis of theca cells. Briefly, cells were washed twice with DPBS and resuspended in Binding Buffer, and 500 μL aliquots were incubated with 6 μL Annexin V-FITC (Sigma-Aldrich) and propidium iodide (10 μL) for 10 min. Fluorescence was measured with a FACSVantage SE flow cytometer (BD Biosciences, Oakville, ON, Canada) and analyzed with Cell Quest Pro software (BD Biosciences).

### 5.8. Statistical Analysis

All statistical analyses were performed with software (JMP) from the SAS institute (Cary, NC, USA). Main effects of treatments were determined by ANOVA, and data were transformed to logarithms if they were not normally distributed (Shapiro–Wilk test). Arcsine transformations were performed on proportional data before analysis. Differences between two means were tested with Student’s *t*-test, and differences between three or more means were established with the Tukey–Kramer HSD test. Experiments were performed with four independent biological replicates. Data are presented as means ± SEM, and *p* < 0.05 was considered statistically significant.

## Figures and Tables

**Figure 1 toxins-15-00228-f001:**
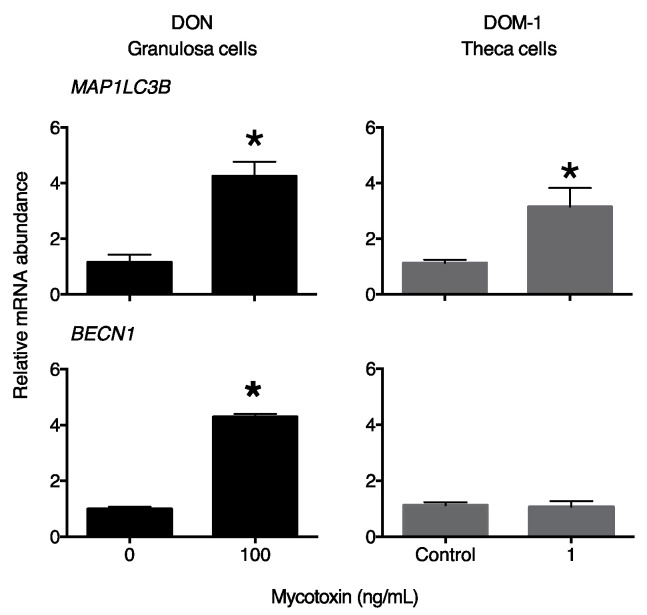
Mycotoxins DON and DOM-1 increase markers of autophagy in bovine granulosa and theca cells, respectively. Cells were placed in serum-free medium and cultured with 100 ng/mL DON (granulosa, black bars) or 1 ng/mL DOM-1 (theca, grey bars) for 4 days. Abundance of LC3 and BECN1 mRNA was measured by qPCR and expressed relative to a control sample. * denotes significant effects of treatment (Student *t*-test, *n* = 4).

**Figure 2 toxins-15-00228-f002:**
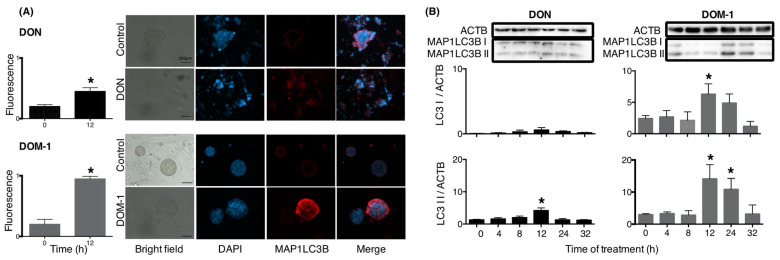
DOM-1 increases cleavage of LC3 in bovine theca cells. (**A**) Cells were placed in serum-free medium and cultured with 100 ng/mL DON (granulosa, black bars) or 1 ng/mL DOM-1 (theca, grey bars) for 12 h and fixed for immunohistochemical staining of LC3 protein. Nuclei were identified with DAPI, and protein abundance is expressed as LC3 fluorescence relative to DAPI fluorescence. The scale bar in the bright field images is 200 μm. (**B**) Cells were cultured with 100 ng/mL DON (granulosa, black bars) or 1 ng/mL DOM-1 (theca, grey bars) for 0–32 h and total (LC3 I) and cleaved (LC3 II) protein abundance measured by immunoblotting. The gel above the graph illustrates a representative experiment with samples loaded in the same order as the bars. * denotes significant effects of treatment ((**A**), Student *t*-test, *n* = 4; (**B**), Tukey–Kramer, *n* = 4) compared to controls without mycotoxin.

**Figure 3 toxins-15-00228-f003:**
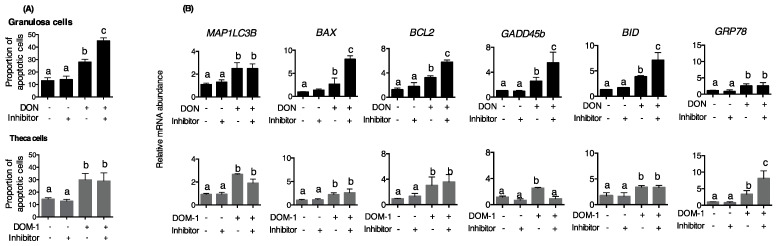
Effect of inhibiting autophagy on mycotoxin-induced cell apoptosis. Cells were placed in serum-free medium and cultured with 100 ng/mL DON (granulosa, black bars) or 1 ng/mL DOM-1 (theca, grey bars) for 5 days before pretreatment with the UKL1 inhibitor XST-14 (5 μM) 60 min prior mycotoxin treatment. Controls received vehicle alone, and cells were harvested at the same as the treatments. (**A**) Flow cytometry was used to measure the proportion of apoptotic cells, and (**B**) autophagy and apoptosis markers were measured by qPCR. Means without common letters are significantly different (Tukey–Kramer HSD test, *n* = 4).

**Figure 4 toxins-15-00228-f004:**
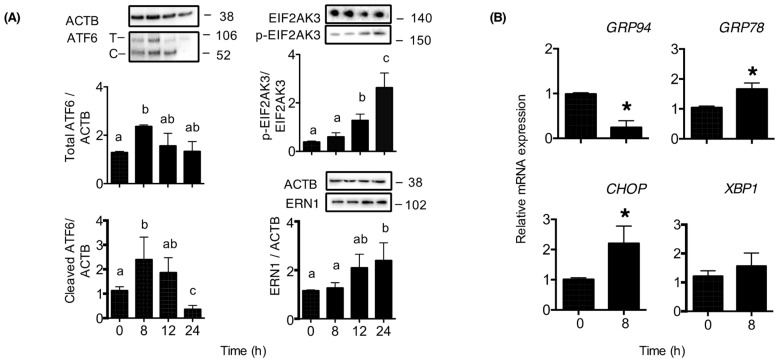
Effect of DON on ER stress pathways in granulosa cells. Cells were placed in serum-free medium and cultured for 5 days. (**A**) Cells were then treated with 100 ng/mL DON and cells were recovered at 0, 8, 12 and 24 h for measurement of abundance of total (T) and cleaved (C) ATF6, total and phosphorylated EIF2AK3 and total ERN1 proteins (immunoblotting). Means without common letters are significantly different (Tukey–Kramer HSD test). (**B**) Cells were treated with 100 ng/mL DON for 0 and 8 h to measure abundance of mRNA encoding key ER stress markers (qPCR) and * denotes significant effect of treatment (Student *t*-test, *n* = 4).

**Figure 5 toxins-15-00228-f005:**
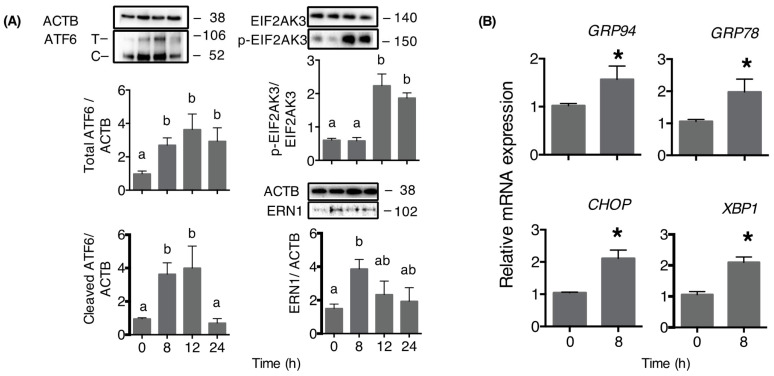
Activation of the ER stress response by DOM-1 in theca cells. Cells were cultured under serum-free conditions for 5 days. (**A**) Cells were then treated with 1 ng/mL DOM-1 and were recovered at 0, 8, 12 and 24 h for measurement of abundance of total (T) and cleaved (C) ATF6, total and phosphorylated EIF2AK3 and total ERN1 (immunoblotting). Means without common letters are significantly different (Tukey–Kramer HSD test). (**B**) Cells were treated with 100 ng/mL DON for 0 and 8 h to measure abundance of mRNA encoding key ER stress markers (qPCR), and * denotes significant effect of treatment (Student *t*-test, *n* = 4).

**Table 1 toxins-15-00228-t001:** Forward (F) and reverse (R) primers used in RT-qPCR.

Gene	Sequence 5′→3′
*GRP78*	F: TGCGAAGCCCTATAGCTGAC R: AGTAGGTGGTACCCAGGTCG
*GRP94*	F: TGCTGTGTGGAGAGGGAATG R: TCCTGTGACCACAATCCCAA
*XBP1*	F: GCA GAG ACC AAG GGG AAT GG R: CTG CAG AGG TGC ACG TAG TC
*MAP1LC3A*	F: CCAGCAAAATCCCGGTGATAA R: TCATGTTGACATGGTCCGGG
*GADD45*	F: TACGAGTCGGCCAAGCTGAT R: GTCCTCCTCTTCCTCGTCGAT
*BAX*	F: AACATGGAGCTGCAGAGGAT R: CAGTTGAAGTTGCCGTCAGA
*BCL2*	F: ATGACTTCTCTCGGCGCTAC R: CTGAAGAGCTCCTCCACCAC
*CHOP*	F: GCACCAAGCATGAACAGTTG R: ATCGATGGTGGTTGGGTATG
*BID*	F: CTCCGTCCTGCTGCTCTTTC R: GTGGACGGCCTTCACCG
*BECN1*	F: CCCAGCTGAAACCAGGAGAG R: GTGGACATCATCCTGGCTGG
*ACTB*	F: GGATGAGGCTCAGAGCAAGAGA R: TCGTCCCAGTTGGTGACGAT
*H2AFZ*	F: GCGGAATTCGAAATGGCTGG R: GGGAAACCGCCTTTGTCTTG

**Table 2 toxins-15-00228-t002:** Antibodies used in Western blot.

Name of Antibody	Manufacturer (Cat. No.)	Dilution	Blocking Solution	Incubation Time
β-actin (C4)	Santa Cruz (sc-47778 HRP, Dellas, TX, USA)	1:5000	Skim milk/TTBS	Overnight
LC3	Novus Biologicals (NB100-2220, Englewood, CO, USA)	1:1000	TTBS	Overnight
EIF2AK3	Cell Signaling (C33E10, Danvers, MA, USA)	1:1000	Skim milk	Overnight
phospho-EIF2AK3	Cell signaling (T980 16F8)	1:1000	Skim milk	Overnight
ATF6α	Santa Cruz (SC-166659)	2:1000	Skim milk	24 h
ERN1	Bioss (SER726, Woburn, MA, USA)	1:1000	Skim milk	Overnight
Anti-mouse IgG, HRP conjugated	Calbiochem (402334, San Diego, CA, USA)	1:10,000	Skim milk/TTBS	1 h
Anti-Rabbit IgG, HRP conjugate	Promega (W401B, Madison, WI, USA)	1:10,000	Skim milk/TTBS	1 h

## Data Availability

Data will be made available upon reasonable request.

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
