# Peer review of "The Mycotoxin De-Epoxy-Deoxynivalenol (DOM-1) Increases Endoplasmic Reticulum Stress in Ovarian Theca Cells"

_toxins, 2023, doi:10.3390/toxins15030228_

Round 1

Reviewer 1 Report

This paper presents an investigation of the mechanisms of action of a feed contaminant and mycotoxin DON and its metabolite DOM-1, in bovine follicular cells (theca and granulosa).

Specifically, whether there was activation of the ribotoxic stress response and/ or endoplasmic reticulum stress pathway in response to DON / DOM-1.

The in vitro experiments utilised IF, FACs, SDS-PAGE western blotting and qPCR to determine changes in cell death and critical markers of each pathway.

The authors conclude that the mechanism of action of DOM-1, at least in bovine theca cells, is through the activation of ER stress.

The paper reads quite well, with few errors of grammar or typos.

The experiments seem appropriate to answer the questions posed.

There are some inconsistencies in the description of data between figure and text which need to be corrected (see below).

Some figures are too small (Fig 2A), and with narrow field of view, it is impossible to judge whether the staining is representative. This needs to be improved.

In addition, please find some minor comments and additional info below: 

1.      Line 24 – what does “consistently reported” mean in this context ?

2.      Line 24-9 – would be interesting to add a little more detail to the on-farm scale of the problem / context. How much of a problem is this mycotoxin in ruminants, is it consistently found but of limited impact etc ?

3.      Line 86 - In granulosa cells, DON increased LC3-II but not LC3-I protein abundance at 12 h of exposure, whereas in theca cells, DOM-1 increased both forms of LC3 protein at 12 and 24 h exposure (Fig2b).

This appears incorrect - There is no significance of LC3I in response to DOM1 at 24 hours.

4.      Fig 2 IF; images are too small and too narrow of field to determine if the responses shown are representative. Plus no scale bar included.

5.      Addition of DOM-1 alone to theca cells increased the abundance of mRNA 109 encoding LC3, BAX, BCL2, GADD45B and BID,

And also GRP78

6.      Fig 4 legend needs to detail cleaved (C) v total (T) ATF6

7.      Fig 4 PERK band is not as clear and sharp as PERK Fig 5; can it be improved ?

8.      Line 130 - Decreased abundance of mRNA encoding GRP78 and GRP94,

This appears incorrect - Not supported by graph (GRP78 increases)

9.      Fig 4b (and Fig 5b), formatting of label has shifted (should read 150 ?)

10.   Line 132 - In theca cells, addition of DOM-1 caused an increase in cleaved ATF6 abundance, and an increase in EIF2AK3 phosphorylation over 12-24 h post-treatment.

As worded, suggests that ATF6 increases over 12-24, but sig diff is rather 8-12.

11.   Is there a reason for using both EIF2AK3 and PERK terms ? Would be more straight forward to be consistent in figure and text describing said figure. Similarly IRE v IRE1a v IREa multiple terms used.

12.   Fig 5 - Is ATF6 decreased at 24h post DOM-1 ?

13.   Line 151 – insert ref.

14.   Line 159 - increased abundance of mRNA of ATF6 target genes, including those 159 encoding the ER-resident proteins GRP78 and GRP94

15.   Line 210 & 223 - 150 mesh steel sieve – should this be 150 micron mesh ?

16.   Table 1 doesn’t include the HKG primers (or give a ref instead) for H2AFZ and beta actin

17.   Ref 31 - reformat

Author Response

We thank the reviewer for their constructive comments which we have answered in the attachment.

Author Response

This review has been submitted by mistake, as the experiments mentioned are not in our manuscript.

Author Response

(The authors gave the same response as above.)
